# NOD1-Targeted Immunonutrition Approaches: On the Way from Disease to Health

**DOI:** 10.3390/biomedicines9050519

**Published:** 2021-05-06

**Authors:** Victoria Fernández-García, Silvia González-Ramos, Paloma Martín-Sanz, José M. Laparra, Lisardo Boscá

**Affiliations:** 1Instituto de Investigaciones Biomédicas Alberto Sols (CSIC-UAM), Arturo Duperier 4, 28029 Madrid, Spain; bvfernandez@iib.uam.es (V.F.-G.); pmartins@iib.uam.es (P.M.-S.); 2Centro de Investigación Biomédica en Red en Enfermedades Cardiovasculares (CIBERCV), Melchor Fernández Almagro 6, 28029 Madrid, Spain; 3Centro de Investigación Biomédica en Red en Enfermedades Hepáticas (CIBERehd), 28029 Madrid, Spain; 4Madrid Institute for Advanced studies in Food (IMDEA Food), Ctra. Cantoblanco 8, 28049 Madrid, Spain; moises.laparra@imdea.org

**Keywords:** NOD1, immunonutrition, lipids, microbiota, exercise, cancer, personalized medicine

## Abstract

Immunonutrition appears as a field with great potential in modern medicine. Since the immune system can trigger serious pathophysiological disorders, it is essential to study and implement a type of nutrition aimed at improving immune system functioning and reinforcing it individually for each patient. In this sense, the nucleotide-binding oligomerization domain-1 (NOD1), one of the members of the pattern recognition receptors (PRRs) family of innate immunity, has been related to numerous pathologies, such as cancer, diabetes, or cardiovascular diseases. NOD1, which is activated by bacterial-derived peptidoglycans, is known to be present in immune cells and to contribute to inflammation and other important pathways, such as fibrosis, upon recognition of its ligands. Since immunonutrition is a significant developing research area with much to discover, we propose NOD1 as a possible target to consider in this field. It is relevant to understand the cellular and molecular mechanisms that modulate the immune system and involve the activation of NOD1 in the context of immunonutrition and associated pathological conditions. Surgical or pharmacological treatments could clearly benefit from the synergy with specific and personalized nutrition that even considers the health status of each subject.

## 1. Immunonutrition at a Glance

Immunity, nutrition, inflammation, infection, and injury are among the topics that are included in the concept of immunonutrition, an interdisciplinary and emerging field with high future projection [1,2]. A well-balanced diet is fundamental for maintaining human health. The nutritional status of each subject and other immunological biomarkers help immunonutrition practice; however, the whole metabolism of the individuals, along with their genetics, their underlying diseases, and even their lifestyles, need to be taken into account. As we will discuss later, there are numerous cases in which immune-modulating nutrition supposes a useful and cutting-edge tool. People with the potential risk of malnutrition (e.g., children or adolescents, elderly, pregnant women, or athletes) and nutrition or patients suffering from immune-related diseases (including allergies and metabolic disorders such as diabetes) will benefit from immunonutrition studies and approaches. This field also allows the analysis of the roles and effects of nutrients (both of conventional origin or obtained from functional foods) and bioactive compounds in the host health and in the immune system. Lifestyle influences, such as diet composition, sedentary or exercise habits, stress, or sleep routines are also evaluated within immunonutrition. Modulation of immune system responses based on nutrient intake contributes to the expression of essential immunoregulatory genes [3,4,5].

Even if it is a well-known fact that the immune system function and development require an adequate nutritional status, the study of the connections between the immune system and nutrition is rather recent. First, research on the nutrition–infection link was advanced: infection influenced the nutritional condition while nutrition determined the host defense extent. This idea was collected in a 1968 *World Health Organization* monograph, before the great development of immunology as a science and the emergence of immune response studies that put it all together. Nutrients (or micronutrients) deficiencies lead to compromised immunity and host defense. The first article on immunonutrition arrived almost 70 years ago, in 1947. Since then, a multitude of scientific publications on the subject has been released [3,6,7].

Interactions between the immune system and nutrients are subjects of growing research interest. After infection, and innate and acquired immune system activation, metabolic changes happen to release nutrients from adipose tissue and muscle ready to be used by immune cells. These nutrients help to repair tissues, modulate cytokine turnout, or protect tissues from harmful effects of free radicals and other oxidants, etc. Therefore, undernutrition or malnutrition, both of them leading to insufficient or inadequate intake of macro- and micronutrients, negatively affects the immune system response. Immune cell functions, phagocytic activity, host defense, complement system, cytokine release, antibody responses, or affinities are among the immune mechanisms impaired after these altered conditions [4,6,8]. However, this perspective based on the essentiality of nutrients or their biochemical role in supporting the function and activity of the immune system is experiencing significant changes toward a more molecular-targeted influence on immunity. For example, the effects derived from the interaction of certain nutrients (i.e., serine-type protease inhibitors) with the innate immune “Toll-like” receptor (TLR)-4 [9,10] or the role and extent that dietary modulation of this receptor can determine selective functional differentiation response(s) of innate immune mediators are receiving increasing interest. This point of view is aligned with the life sciences-based *Process for the Assessment of Scientific Support for Claims on Foods* (PASSCLAIM) that was coordinated by the Institute of Life Sciences (ILSI) Europe [11].

There are three main and potential targets for immunonutrition: mucosal barrier functionality, cellular defense, and inflammation (local or systemic; related to inflammatory mediators). Antioxidants, vitamin D, fatty acids, carbohydrates, bovine colostrum, prebiotics, probiotics, proteins, minerals, and herbal supplements are some topics associated with immunonutrition. They are called *immunonutrients*, that is, molecular compounds with a double function: they act as components of the diet and at the same time influence the immune system. Arginine, glutamine, n-3 fatty acids, branched-chain amino acids, and nucleotides are the most studied nutrients in this science. Indeed, there exist commercial enteral feeds, which combine some or all of these nutrients. Therefore, two tactics that will be discussed later may govern clinical trials: those that obtain benefit from a single immunonutrient approach or the ones that use a nutrient-combination strategy. In the first case, enteral glutamine supplementation has been demonstrated to reduce the incidence of pneumonia, bacteremia, and fatal sepsis in critically ill patients and the incidence of severe sepsis in premature neonates [12,13]. Glutamine is an essential nutrient for gut mucosal cells and even more necessary if we consider that mucosal atrophy implies bacterial invasion and the entrance of microorganisms into the bloodstream. Accordingly, parenteral glutamine has also been used in clinical trials. Supplemental oxygen requirement, ventilation support, length, and time of stay in the intensive care unit were diminished after enteral lipid-modified feed in moderate and severe acute respiratory distress syndrome patients. In addition, their capacity to develop organ failure and their mortality tended to be decreased [1,5,6,12,14,15].

An inflammatory storm involving immune system activation, pro- and anti-inflammatory mediators, paracrine and endocrine effectors, free radicals and oxidants release lead to multiple organ failure mentioned above in critically ill patients [16,17]. The need for an exogenous supply of specific nutrients in critically ill or surgical patients paved the way for immunonutrition attempts (Figure 1). It should be noted that immunosuppression is a critical period after major surgery (or after hyperinflammation and an excessive compensatory response in critically ill subjects) that exhibits an increased risk of infection, morbidity, and mortality. This area is where immunonutrition comes into play, strengthening the immune system and favoring selectively driven response(s) to reduce derived complications and improve the clinical evolution of these patients. Early treatment is key to reduce inflammation and avoid/decrease the immunosuppression phase. However, there is much controversy about the real benefits of immunonutrition in critically ill patients; therefore, more studies are needed [1,2,12,15,18].

Exercise practice can benefit greatly from immunomodulatory nutrition protocols. Experts in the field have elaborated a large document that contains scientific evidence, critical points of view, and new perspectives in this field (*Consensus Statement. Immunonutrition and Exercise* [7]).

Since immunonutrition has emerged as a potential ally to fight numerous diseases or as a way to improve human health in several circumstances, different related societies have been founded. Among these institutions, it is worth mentioning the International Society for Immunonutrition (ISIN), due to its ability to establish itself worldwide. The latest example of this important contribution is its aim of promoting adequate nutrition to strengthen the immune system and of establishing strategies of *trained innate immunity*, as well as, especially acquired immunity, against COVID-19 infection. Based on scientific evidence and promoting the expansion and dissemination of knowledge in this field of nutrition linked to immunity, these societies contribute to the development of immunomodulatory nutrition (https://immunonutrition-isin.org/ accessed on 30 March 2020).

There are different parameters and measurements that can help studies on immunonutrition and the application of new therapeutic strategies. Several groups have analyzed markers for their suitability to indicate the modifications that nutrients exert on the immune system. For example, immune responses can be modified by decreased cell proliferation, decreased protein synthesis, nutrient deficiencies, and impaired metabolic pathways under malnutrition conditions. Other useful immunological biomarkers and immunonutrition evaluation parameters are the following: circulating factors such as immunoglobulins in human fluids (IgG, IgA, IgM, IgD, and IgE); acute phase proteins (complement factors, C-reactive protein, and ceruloplasmin); levels of cytokines and cytokine receptors; immunocompetent cell count and functionality, and other functional and genetics of the immune system [3,4].

The immune system is a highly complex defense network of mechanisms, signals, cells, organs, and tissues, whose goal is to protect the human being. Conventional versus nonconventional or extramedullary hematopoiesis are key processes by which blood cellular components (including the most relevant immune actors, leukocytes) are formed [19]. However, this intricate system is extremely variable between subjects and can also cause illness when it fails or there are deficiencies in its performance. Many immunometabolic-based chronic diseases such as cancer, the nonalcoholic fatty liver disease that carries a high risk of type 2 diabetes, obesity, and other important features of metabolic syndrome, Alzheimer’s disease, and autoimmune or cardiovascular diseases have been related to inappropriate immune and inflammatory responses. However, both hereditary and nonhereditary determinants modulate the variability of the immune system [20]. Therefore, an in-depth study and understanding of these biological processes, in addition to the individualized use of nutrients as regulators of the immune function, is essential.

## 2. State of the Art: Immune Cells and NOD1

Potentially harmful agents are recognized and fought off by the immune system, which responds to countless antigens. Its capacity to distinguish between self or nonself molecules and events and to establish tolerance of self-elements is essential for the subject. This includes external components and beneficial commensal microbiota. There are three main levels of immune defense: (1) physical (e.g., skin, mucous membranes, endothelia), physiological (such as sneezing or diarrhea), chemical (e.g., low pH and antimicrobial molecules), or biological (normal microbiota) barriers (all of them normally known as physicochemical barriers); (2) innate or unspecific immunity (for a rapid defense action); (3) acquired, adaptive or specific immunity (more complex and persistent, leading to immunological memory). Immune responses comprise different blood-borne issues, including soluble components and cells.

The unspecific immunity involves soluble factors (cytokines, complement factors, and acute-phase proteins), and cells (granulocytes (neutrophils, basophils, and eosinophils), monocytes/macrophages, and natural killer (NK) cells). Adaptive immunity implies a more efficient response after reexposure to a specific antigen. It is executed mainly by B, T, T cytotoxic, and T helper lymphocytes and soluble components (cytokines and antibodies). However, innate and acquired immune systems are interconnected and have been shown to be linked by cell-to-cell or cell–surface protein contacts and by cytokines release. What is clear is that there is certainly a close interaction between both types of immune responses [3,6,21].

Immune cells express pattern-recognition receptors (PRRs) that recognize and are activated by molecular patterns associated with microbes or pathogens (MAMPs or PAMPs, respectively). Different signaling pathways and cell responses are triggered by these receptors [22,23,24,25]. Interactions of PRRs with normal intestinal microbial or nonpathogenic molecules confer a protective mechanism for the host that contributes to immune memory training. Impairments in these mechanisms lead to the development of immune diseases. An essential component of the bacterial cell wall formed by two β-1,4-linked sugars, *N*-acetylglucosamine (NAG) and *N*-acetylmuramic acid (NAM) is called peptidoglycan (PGN) or murein. PGN glycan strands are cross-linked by short peptides (two to five amino acids). The general backbone presents certain diversity when comparing Gram-negative and Gram-positive bacteria. In the last case, L-lysine is usually the dibasic amino acid, while Gram-negative bacteria contain *meso*-diaminopimelic acid (DAP). Among the intracellular cytoplasmic PRRs, nucleotide-binding oligomerization domain-1 and -2 (NOD1/NOD2) share overlapping signaling pathways but different specificities in MAMPs/PAMPs recognition [26,27,28,29,30,31]. NOD1 recognizes and is activated by bacterial PGNs. To this purpose, the γ-D-glutamyl-DAP (iE-DAP) is sufficient to activate NOD1. iE-DAP is mainly associated with Gram-negative bacteria, but it is also present in a few Gram-positive bacteria (e.g., *Listeria monocytogenes* or *Bacillus spp*.). Nevertheless, Gram-negative bacteria constitute the main class of microorganisms present in the gastrointestinal microbiota. These changes in bacterial composition can have important consequences on disease severity since PAMPs from the gut microbiota modulate intestinal mucosal immunity via TLRs and other PRRs. Gut microbiota composition has been directly associated with innate immunity [32]. In addition, changes in the diversity of the intestinal microbiota can worsen or improve insulin resistance and therefore type 2 diabetes (T2D) in mice and patients [33,34], possibly due to interaction with innate immune signaling through the TLR4, NOD1/NOD2 proteins, etc., as yet undefined pathways [35,36]. Activated NOD1/NOD2 recruit receptor-interacting serine/threonine-protein kinase 2 (RIPK2), transducing the activation of the “*nodosome*” after RIPK2 autophosphorylation (Figure 2). RIPK2 is further polyubiquitinated by E3 ubiquitin ligases, which allow the additional recruitment of mitogen-activated protein kinase kinase kinase 7 (MAP3K7, also known as TAK1) and subsequent activation of the IKK/NF-κB signaling pathway [31,37,38,39,40].

NOD1 is expressed in several cells, from nonhematopoietic (such as endothelial cells) to hematopoietic and immune cells (e.g., monocytes/macrophages, neutrophils, NK cells, lymphocytes). Functional NOD1 in neutrophils, for example, has been related to *Staphylococcus aureus* and *Streptococcus pneumoniae* clearance [41,42,43,44]. However, this PRR is associated with innate immunity and with the acquired immune response (Figure 2). For example, NOD1 stimulation primes antigen-specific T cell immune responses primarily with a Th2 polarization profile. Interestingly, along with other TLRs of innate immunity, NOD1 leads to Th1, Th2, and Th17 immune responses. Furthermore, NOD1 activation and downstream signaling contribute to B cell antigen receptor-engaged mature B cells’ survival [45,46].

Regarding immunonutrition, both counts and functionality of phagocytic neutrophils and monocytes/macrophages are important due to their response to infections and to their involvement in autoimmune diseases. From an immunonutritional point of view, the myeloid F4/80 population in rodents and the epidermal growth factor module-containing mucin-like receptor 1 (EMR1) equivalent in humans appear to be susceptible to the immunonutritional-driven polarization by dietary protease inhibitors [47,48]. Furthermore, when considering immunonutritional strategies to include protease inhibitors in complex food matrices (e.g., bread formulations), they have been shown to be effective in controlling insulin resistance [49]. The normal reference values are relevant for the practice of immunonutrition, especially when there is malnutrition or special pathologies. Nutritional status can be indirectly measured by total lymphocyte count, establishing a gradient from mild to severe malnutrition based on their number of cells per mm^3^ or by immunocompetent phenotypic and functional flow cytometry assays [3].

## 3. NOD1-Related Diseases and Nutrients: The Potential Benefit of Immunonutrition Approaches

There are at least 196 reported diseases associated with NOD1, from several cancers or neoplasms to inflammatory, metabolic, immune, and infectious diseases, as shown in the *Open Targets Platform*: https://www.targetvalidation.org/target/ENSG00000106100/associations (accessed on 20 April 2021). Since NOD1 was discovered, it has always been considered an active and key mediator in human metabolism and inflammation. It is directly involved in signaling pathways of cell stressors (e.g., calcium influx or endoplasmic reticulum mechanisms, as indicated later) and regulates downstream pathways in response to metabolic mediators such as glucose and fatty acids. NOD1 stimulation is believed to be the result of a combination of bacterial and metabolic ligands [50,51].

There exists a link between immunocompetent cell functions and adipose tissue homeostasis, including leptin and TNF-α in adipose tissue [7,8]. In this line, cell death plays also a determinant role within these mechanisms [52]. Programmed cell death and innate immune system signaling are closely related. NOD1 activation has been connected with cell death induction (mainly via caspase 8 activation) and, in addition, cell death is involved in the regulation of inflammatory processes [53,54,55]. Inflammation was shown to be partially modulated by certain nutrients such as fatty acids, retinol, tocopherol, or zinc. This observation established a new interesting field of study within immunonutrition where balancing n-3 and n-6 fatty acid consumption was a key intervention [3].

### 3.1. NOD1, Adipose Tissue, and Obesity

Consumption of Western diet in humans and its equivalent in mice, high-fat diet (HFD), results in the upregulation of NOD1 signaling. These diets involve weight gain, an increase in adiposity, and the development of a proinflammatory onset. Consequently, NOD1 activation in adipocytes leads to NF-κB and protein kinase A-triggered lipolysis. This lipolysis prompts an increase in available fatty acids, which can ultimately lead to insulin resistance. The metabolic effects derived from fatty acids are the consequence of several molecular and cellular mechanisms: from stimulation of hormone receptors to modifications in the fluidity of the cell membrane. In this regard, saturated fatty acids (e.g., lauric acid) favor NOD1 activation in several cell types, including adipocytes, while unsaturated fatty acids (e.g., docosahexaenoic acid (DHA)) impair NOD1 auto-oligomerization and its ligand-induced inflammation. In fact, DHA negatively regulates the stimulation of NOD1 signaling by saturated fatty acids. Therefore, nutritional interventions targeting fat and the immune system at the same time would be promising. Fat-enriched diets are well-known drivers of metabolic syndrome and cardiovascular diseases [50,56]. In this context, the administration of immunonutritional protease inhibitors has been shown to be effective in modulating lipid homeostasis and shaping the gut microbiota, ameliorating the severity of hepatocarcinoma under HFD [10].

### 3.2. NOD1 and Atherogenesis

Our group and others [57,58,59,60] have described a prominent role for NOD1 in the leading cause of death worldwide: cardiovascular diseases (CVDs). NOD1 has been shown to contribute to atherogenesis, both in early and advanced disease. Since atherosclerosis is the main underlying cause of most of the CVDs and it involves lipid deposition and NOD1 in addition to other important inflammatory and immune components, immunonutrition approaches represent another essential front to fight them. NOD1 is induced and activated in atherosclerotic tissues. NOD1 deletion in *Apoe^−/−^* mice, one of the most commonly used animal models of atherogenesis, reduces the burden of the disease and the accumulation of leukocytes within the lesions, especially monocytes and neutrophils. Endothelial NOD1 activated by PGNs or oxidized LDLs (oxLDLs, important within diet and lipid sources), leads to overexpression of VCAM-1 and subsequent adhesion of leukocytes to athero-prone vessels [57,58,59,60]. When atherogenesis progresses, there is a risk of atherothrombosis and acute coronary syndromes, triggered by unstable plaque rupture, cell necrosis, hyperinflammation, and decreased collagen content. In these circumstances, cells such as macrophages and smooth muscle cells play a critical role since they proliferate, undergo apoptosis, and form foam cells within the plaque. NOD1 is involved in all these processes that contribute to the vulnerability of atheromas [58,61]. Cardiac dysfunction, impaired excitation–contraction coupling, and Ca^2+^ handling have also been associated with NOD1 [62]. Furthermore, subsequent studies have shown that NOD1 activation by noxious stimuli and stress is also involved in cardiac remodeling and hypertrophy, heart failure, and mitochondrial energy balance and mitophagy [63]. Therefore, knowing that the diet and its derived nutrients and metabolites influence both the energy state of the cells and the production of damage signals, immunomodulatory nutrition strategies become important.

### 3.3. NOD1 and Diabetes

Consistent with this, it has also been observed that mice fed HFD have higher glucose and insulin basal concentrations, insulin resistance, and glucose intolerance. Not only high-fat diets positively regulate NOD1, but also high sugar intake, as occurs in the Western diet [56,64,65,66,67,68,69,70,71,72]. For example, high glucose levels can activate NOD1 in mesangial cells. In fact, adipocytes stimulated with iE-DAP or other NOD1 ligands exhibit enhanced pro-inflammatory responses and diminished insulin receptor signaling, leading to a decreased glucose uptake by insulin and to insulin resistance [50]. In addition, diabetes, obesity, and hyperglycemia cause dysbiosis of the intestinal microbiota favoring gut permeability and the consequent increase in circulating bacterial products. Activation of NOD1 by these microbe-derived molecules can disrupt other signaling pathways and metabolic routes, thus establishing a self-feedback loop [50,73]. For example, microbiota recognition by NOD1 in bone marrow mesenchymal stromal cells can modulate hematopoiesis in mice [74]. In contrast, NOD1-mediated renal antibacterial defenses are dampened by cyclosporin A (CsA) [75]. Another NOD1–microbiota–metabolism connection is shown in a work by Zhang et al., in which they demonstrate that microbes-derived NOD1 ligands directly regulate insulin trafficking in pancreatic beta cells. NOD1 ligands are released into the bloodstream by intestinal lysozyme and Rab1a, recruited by NOD1, leading to insulin signaling [76]. What is clear is the importance of micro and macronutrient homeostasis in this immune system–microbiome–metabolism axis. Interestingly, evidence has shown that vitamin D administration downregulates 66 genes and upregulates 291 genes involved in the biological function and homeostasis of more than 160 signaling pathways associated with metabolic dysfunctions, CVDs, autoimmune disorders, and cancer [4].

### 3.4. NOD1 and Cancer

The NOD1–cancer axis has been widely confirmed and it is quite relevant due to the high morbidity and mortality associated with these diseases [25,39,54,77,78,79,80,81,82]. For instance, a recent study has shown that activated NOD1 leads to the progression of colorectal carcinogenesis by modulating the immunosuppressive functions of tumor-infiltrating leukocytes via arginase-1 activity [77]. Other previous studies have also linked NOD1 activities to colon cancer and metastasis [79,82] and carcinogenic responses to *Helicobacter pylori* pathogen [80]. Given the high impact of nutrition on the gastrointestinal tract and the microbiota content (also closely linked to NOD1) and the information provided by these studies, new immunonutrition strategies are required. There are also studies on the involvement of NOD1 in other tumor-related diseases such as cervical cancer, breast cancer, and head and neck squamous cell carcinoma [25,39,54,78,81,83]. Patients with these cancers would benefit from immunomodulatory nutrition strategies targeting NOD1.

### 3.5. NOD1 and Kidney Dysfunction

Clear cell renal cell carcinoma is another example of the involvement of NOD1 in cancer processes, although it is related to the pathogenesis of another organ—the kidney [84]. NOD1 is expressed in human and murine renal tubular epithelial cells [85,86]. Renal ischemia/reperfusion injury activates NOD1 due to tubular epithelial apoptosis and inflammation [85,87]. Furthermore, a complex and interesting study linked immunosuppressive drugs such as CsA with NOD1: CsA inhibited the innate renal antibacterial responses of NOD1 in both mice and human transplant recipients [75]. Other NOD1/NOD2 KO models have been used to decipher the role of these PRRs in the onset of renal disease and inflammation [88]. Due to all these data and the fact that kidneys are involved in nutrients and xenobiotics metabolism, as well as in blood filtering, a combined NOD1-immunonutrition therapy must be tackled in specific patients. In addition, high-protein diets can damage the kidneys and lead to severe diseases, which is why immunomodulatory nutrition is essential.

### 3.6. NOD1 and Respiratory and Hepatic Diseases

Lung cells, including epithelial cells, are also involved in the activation and functions of NOD1. As an example, NOD1 activates CCAAT/enhancer-binding protein β (C/EBPβ) signaling pathways in these cells after bacterial or agonists recognition [89]. Since lung and respiratory tract diseases could benefit from immunonutrition, as shown in the case of acute respiratory distress syndrome, NOD1 inhibition or downregulation appears as a potential therapeutic target [90].

NOD1 regulates liver molecular and cellular mechanisms as well. For example, this immune PRR contributes to liver ischemia/reperfusion injury. NOD1 modulates the overexpression of adhesion molecules and regulates infiltration of neutrophils in the liver, which are responsible for causing the injury [91]. Given that the liver constitutes a “metabolic turbine” where multiple anabolic and catabolic processes occur and where nutrients are metabolized, immunonutrition and NOD1-targeted approaches are likely candidates to restore liver homeostasis and function.

### 3.7. NOD1 and Thyroid Hormones Homeostasis

Recent work published by our group links NOD1 with metabolism, thyroid hormones, and energy expenditure. NOD1 is also expressed in the thyroid and other highly metabolic cells and tissues, such as adipose tissue, including inguinal white adipose tissue (iWAT) and epididymal white adipose tissue (eWAT). NOD1 activation in Wt mice after 6 weeks HFD regimen resulted in decreased lipogenesis in iWAT and eWAT, compared to their *Nod1^−/−^* counterparts, as well as differences in bile acid metabolism and thyroid hormone homeostasis. Interestingly, *Nod1^−/−^* mice showed accelerated diet-induced obesity, demonstrating the involvement of this PRR in metabolism and body weight regulation and becoming a good target for immunonutrition. Accordingly, NOD1 influences thyroid hormones’ functions (T3, T4), which are implicated in the modulation of metabolic routes related to lipids (e.g., lipogenesis and lipolysis), carbohydrates, bile acids, energy storage/expenditure, and thermogenesis. Furthermore, all of these complex processes and their interrelationships can lead to severe disorders, such as hypothyroidism, hyperthyroidism, Graves’ disease, or even cancer.

The biology of the thyroid is also linked to infections and sepsis. Along these lines, it has been confirmed that the intestinal microbiota is altered depending on HFD and NOD1 function. The immune system has a relevant role in this regard, and we report that immune cells express thyroid receptors and these cells are modulated directly or indirectly by thyroid hormones. In addition, the thyroid gland is regulated by the nutritional status and other signals such as leptin or appetite-controlling peptides. Therefore, there is a need for an immunonutrition approach for thyroid dysfunctions involving or not involving NOD1 [41,58].

### 3.8. NOD1, Thyroid Hormones, and Pregnancy

Thyroid hormones and the endocrine system are also crucial during pregnancy for normal fetal and neonatal brain development. Interestingly, several NOD1–pregnancy relationships have been established. NOD1 is widely expressed in maternal and fetal tissues. For example, NOD1 in the uterine wall, decidua, and placenta causes inflammation in both normal and preeclamptic pregnancies, but especially in the first case, as it is upregulated [92]. Furthermore, NOD1 regulates the decidual stromal cell role in the maintenance of pregnancy during the first trimester [70]. NOD1 expression and activation have also been investigated in the fetal membrane, placenta, and plasma obtained from pregnancies with premature rupture of membranes [93]. Accordingly, related to the previous comments on NOD1 activation in conditions of high glucose and diabetes, and its relationship with insulin resistance, this PRR expression is enhanced in the adipose tissue of women with gestational diabetes. Notably, increased maternal inflammation and peripheral insulin resistance consist of two features of pregnancy, which evolved with gestational diabetes mellitus [94]. Other works linked NOD1 activation and signaling with fetal death and growth restriction through fetal–maternal vasculopathy, thus connecting several pathogenic events [95]. Furthermore, after *Chlamydia trachomatis* infection, NOD1 participates in the induction of IL-1β release in human trophoblasts [96], cells in which the regulation of NOD1 and the pattern of cytokine secretion have been analyzed [97]. Prematurity has been strongly associated with infection, and there are studies that go further and relate it to the activation of NOD1 by bacterial ligands sensing [98]. Additionally, pregnancy constitutes a physiological context that exemplifies the importance of immunometabolic interactions for both the mother and the embryo. This is a unique physiological state, in which the immune tolerance of the developing fetus coincides with the immune activation to induce insulin resistance necessary to meet the energy needs of the mother, along with an increased demand for glucose by the embryo [99]. The intergenerational transfer of immunometabolic properties between mother and offspring can help prevent further development of chronic diseases as well as their severity [100]. Given the importance of balanced nutrition in pregnant women and the connections between them, the immune system, and the implications of NOD1 for health and disease, immunonutrition seems such as a perfect treatment combination approach.

NOD1 participates in processes related to lactation. This occurs even in other species such as cows or goats. In dairy cows, for example, a high-concentrate diet-induced apoptosis of mammary epithelial cells by the NOD1 and caspase-8 pathways [101] while triggering NOD1-NF-κB downstream proinflammatory signaling pathways [102]. Another interesting study cited the relationship between NOD1 downregulation in neutrophils from periparturient dairy cows, the dysregulation of neutrophils, and their increased vulnerability to infectious diseases [103]. In goats, sodium butyrate inhibited NOD1 activation in lactating mammary glands during subacute ruminal acidosis. NOD1 signaling pathway activation in cow and goat mammary glands usually occurs during ruminal acidosis after translocation of lipopolysaccharide from the digestive tract into the bloodstream [101]. Therefore, studies on these issues suggest two ways of modulating NOD1 under nutritional conditions: acting indirectly on livestock (which provide us with food and factors that regulate our immune system) or studying these mechanisms in humans and applying them to improve immunity.

### 3.9. NOD1 and Inflammatory Diseases

An adequate nutritional status is required for optimal wound healing, thus setting a promising outlook for immunonutrition. Molecular mechanisms such as inflammation, cell migration or proliferation, coagulation, and remodeling, contribute to wound healing. Therefore, nutritional depletion or deficiency of certain nutrients could lead to an impaired wound healing process. In fact, malnourished subjects often have ulcers, infections, and delayed healing or even nonhealing wounds. In contrast, nutritional supplementation with beneficial effectors, such as arginine or curcumin, stimulates restorative events in this host [104,105,106,107]. Given that migratory leukocytes and immune responses play a fundamental role in these processes and that NOD1 is an active actor in immune cells, it would not be surprising if combined strategies based on wound healing favored by this receptor were taken into account.

Another example of the contribution of NOD1 to human disease is illustrated in the expression and functions of NOD1 in dental pulp odontoblasts. NOD1 activation and derived release of chemokines lead to the appearance of pulpitis. This is interesting because odontoblasts recognize caries-related pathogens and trigger inflammation and subsequent pulpitis [108]. In this regard, NOD1 was demonstrated to sense periodontal pathogens (e.g., *Porphyromonas gingivalis*, *Fusobacterium nucleatum*, and *Aggregatibacter actinomycetemcomitans*) [109] and putative newly identified pathogens (*Eubacterium nodatum*, *Eubacterium saphenum*, and *Filifactor alocis*) to initiate innate immune responses [110]. This *spp*. are important in the development of periodontitis inflammatory disease. Interestingly, other external factors, such as cigarette smoke, have been determined to regulate NOD1 signaling and β-defensins in the oral mucosa and therefore host defense against pathogens [111]. A carefully studied immune-related nutrition can help both prevent tooth decay and disrupt NOD1 activation (Figure 3).

## 4. A Focus on Future Strategies

It is widely shown that human nutrition is associated with the content and function of the gut microbiota. In turn, the intestinal microbiota is related to metabolic diseases and the immune system. The prevalence of certain diseases, such as metabolic ones, including obesity, has been increasing over the past decades [112]. In an interesting study, exercise was opposed to obesity-related changes in the gut microbiota [113]. In another work, scientists highlighted the importance of combining novel diet and gut microbiome-modulating strategies to treat inflammatory bowel disease in an immunonutrition approach [114]. Therefore, there are strong connections between diet, exercise, microbiota, immune system, and health or disease that highlight the need to understand better the mechanisms involved in the concept of immunonutrition. A pronounced leukocytosis, including a high count of neutrophils, results after harsh exercise. In addition, immunosuppression induced by a prolonged or exhaustive exercise triggers dysfunction of the immune system and inflammatory responses. These disorders eventually lead to a pronounced decrease in the resistance to pathogens and to an overall increased risk of infections and diseases. Some biomarkers of immune function, such as the number and activity of immune cells, the expression of the major histocompatibility complex II, the delayed-type hypersensitivity response, salivary IgA, etc., are altered after intense exercise [7,14,115].

Glucose, amino acids, and lipids are nutrients that have a clear function in immunonutrition, contributing to energy production in immune cells and to their correct proliferation. Micronutrients, such as zinc, iron, magnesium, and vitamins, are essential for proper and competent immune function. It is noteworthy that some nutrients or micronutrients must be obtained from the diet since they cannot be directly synthesized in human cells or cannot be sufficiently produced. Novel immunonutrition supplements such as β-glucan, quercetin, or curcumin hold promise for improving immune function and reducing infections after vigorous exercise [7,13,14,115]. Recently published work has identified a specialized bone marrow cell progenitor that contributes to lymphocyte production in response to exercise, suggesting that mechanosensitive osteogenic progenitors are involved in immune system responses and in the clearance of bacterial infections [116]. Regarding NOD1, even if this PRR and exercise training have not yet been directly related, it is clear that NOD1 activation contributes to insulin resistance, cardiac dysfunction, and muscle contraction [62,68].

As mentioned above, critically ill patients are an important population that will potentially benefit from immunonutrition strategies [2,5,6,12,14,15,18]. Evidence on the host immune system and gut microbiome interactions in critically ill patients highlights key research areas for the design and use of specific and individualized treatments [16]. Furthermore, NOD1 mRNA is suppressed in monocytes of septic patients, increasing the risk of infections [117]. Accordingly, the expression of genes involved in the innate immune response, including NOD1, changes in critically ill patients suffering sepsis and related malnutrition [17].

Nutritional interventions are considered promising ways to improve impaired immune function and predisposition to infection with aging [118,119]. The elderly, similar to overweight and obese individuals, present an increased inflammatory condition. They generally have a high prevalence of nutrient deficiencies that affect their immune response, defense against infection, and cognitive function. Adapted physical activity training programs are beneficial for this population. Immunosenescence, a cell-mediated reduction in immune function, favors the highest rates of morbidity and mortality in the elderly. However, age appears to be related to an enhanced inflammatory response. Therefore, regular moderate exercise improves immune function in this population. For example, calisthenics exercise stimulates the activities of T lymphocytes and natural killers in older women. Micronutrient deficiency, inflammation, muscle damage, and oxidative stress increased in the elderly, while glucose tolerance and insulin sensitivity decreased with aging. These alterations highlight the need for immunonutritional strategies for them, and if they consider the role of NOD1 in aging, they will be more complete and effective [7,8].

The influence of microRNAs (miRNAs) as key regulatory elements in physiological processes, inflammation, the activity of the immune system, and various human disorders, such as cancer or infectious diseases, has long been known. miRNAs are short, noncoding RNAs that bind to specific mRNA sequences and lead to the suppression or degradation of translation of the respective target mRNAs. They are important posttranscriptional modulators of hematopoietic cell fate decisions, for example, or are involved in the regulation of gene expression associated with the immune response. However, the precise mechanisms involved in these biological and pathological processes are poorly defined in this still developing field [79,120,121,122]. An example of miRNAs in immune-related metabolic diseases is presented in a study in which miRNA-495 inhibited the inflammation caused by high glucose, the extracellular matrix accumulation, and the differentiation of cardiac fibroblasts by NOD1 downregulation [123]. Knowing that NOD1, as we mentioned above, is closely related to atherosclerosis and CVDs, a combined immunonutrition–microRNA strategy would be expected to be effective.

Inhibitors of the immune system and immune checkpoint inhibitors are novel molecules with immunoregulatory effects in different pathologies among which several cancers stand out [124,125]. Some examples to mention that are related to diseases in which NOD1 contributes in some way would be the following referenced works: [48], [126], and [51]. Hence, a double immunonutrition–inhibitor(s) strategy (and even more, targeting NOD1) will be helpful in combating certain diseases.

The process of elucidating antigen receptor heterogeneity and cellular, antibody, and cytokine responses involved in health and disease is called immunoprofiling. This field allows the identification of new biomarkers aimed at determining the severity, course, and outcome of the disease, as well as the patient’s response to a specific treatment. Potential translational applications are immunotherapy of cancer, inflammatory dysfunctions, autoimmune disorders, and infectious diseases. However, with respect to all types of diseases, since *immune cells are almost as diverse as people*, promising and innovative methods of single-cell analysis, combined with immunoprofiling techniques, are being used. Immunoprofiling approaches used in the service of immunonutrition may be of great use in the future [127,128,129].

It is known that immunonutrition studies use vaccination responses to obtain data on immunomodulatory mechanisms. These responses are influenced by factors as wide as environmental, infectious, and related to lifestyle or nutrition. Additionally, peptides with DAP, therefore NOD1 ligands, have adjuvant activity. Of note, Freund’s adjuvant activity requires monomeric PGN subunits as minimal structures to be functional. These NOD1 agonists with adjuvant capacity showed antibacterial and antiviral activities while having antitumor roles. Therefore, there is an obvious link between the immune system, nutrition, vaccination, NOD1, and infection that needs to be investigated in depth. [3,130].

Interestingly, there are different current clinical trials on immunonutrition. Many of them use different nutrient combinations and strategies, such as a mix of arginine, n-3 fatty acids, and nucleotides with/without glutamine in critically ill and surgical patients. Normally, they show higher total nitrogen content, antioxidant vitamins (e.g., A and E), and minerals (such as selenium) levels. Here, is a reduced selection of some of these promising clinical trials: “Immunonutrition Supplementation for Improved Burn Wound Healing in Older Adults” (NCT04725071), “Effect of Preoperative Immunonutrition in Upper Digestive Tract” (NCT04027088), “Preoperative Immunonutrition and Cardiac Patients” (NCT03445221), “Effect of Perioperative Immunonutrition on Recurrence and Infections in Crohn’s Disease Patients (EPIRIC)” (NCT04014517), “Immunonutrition and Carbohydrate Loading Strategies in Breast Reconstruction” (NCT03764943), “Perioperative Immunonutrition, Phagocytic, and Bactericidal Activity of Blood Platelets in Gastric Cancer Patients” (NCT01704664); and others than have been completed: “Changes in Inflammatory Response after Immunonutrition Compared to Standard Nutrition in Colorectal Cancer Tissue” (NCT04732442), “Effect of Enteral Immunonutrition on Immune, Inflammatory Markers, and Nutritional Status in Patients Undergoing Gastrectomy for Gastric Cancer” (NCT03730545), “Preoperative Immunonutrition in Normo-Nourished Patients Undergoing Fast-Track Laparoscopic Colorectal Surgery” (NCT04692545), “Effect of Immunonutrition on Inflammatory Markers after Bariatric Surgery” (NCT03010280), “Effects of Immunonutrition on Biomarkers in Traumatic Brain Injury” (NCT03166449), and “Immunonutrition in Cardiac Surgery” (NCT00247793). All of them prevent or treat highly prevalent human diseases, demonstrating the importance of such novel interventions and the need for more studies. However, none of these trials appear to address a NOD1-related/targeted intervention, which would be of interest (Figure 4).

Thus, NOD1 has great therapeutic potential as a target in several human disorders. The development of new treatments is essential, complementing existing therapeutic and nutritional approaches or even favoring patients who do not respond adequately to existing drugs [130].

## 5. Conclusions

In recent years, immunonutrition has become a broad and hopeful field with promising applications in many diseases. This is the art of modulating the immune system by employing specific nutrient-based interventions. Imbalances in normal metabolism or deficiencies in immunity can contribute to disease development and even to mortality. Since NOD1, an immunity-related receptor, has been shown to be involved in several different disorders and pathologies, research on the combination of immunonutrition, NOD1-targeting, and other translational biomedical fields could benefit low-grade inflammation chronic patients. However, despite the direct relationship of NOD1 with different immunometabolic chronic diseases and cancer, it is still almost completely unknown today in relation to its potential modulation through immunonutritional strategies as aids to biomedical interventions. Further efforts should aim to better understand immunity-mediated diseases while focusing on defining the most effective immunity-modulating nutrition approaches (both individual and approaches in which different immunonutrients are combined) and even their use in conjunction with other types of treatments, surgeries, or lifestyle factors. Given the breadth of the areas covered in this field, what is clear is that deeper interdisciplinary research and collaboration between trained experts are essential.

## Figures and Tables

**Figure 1 biomedicines-09-00519-f001:**
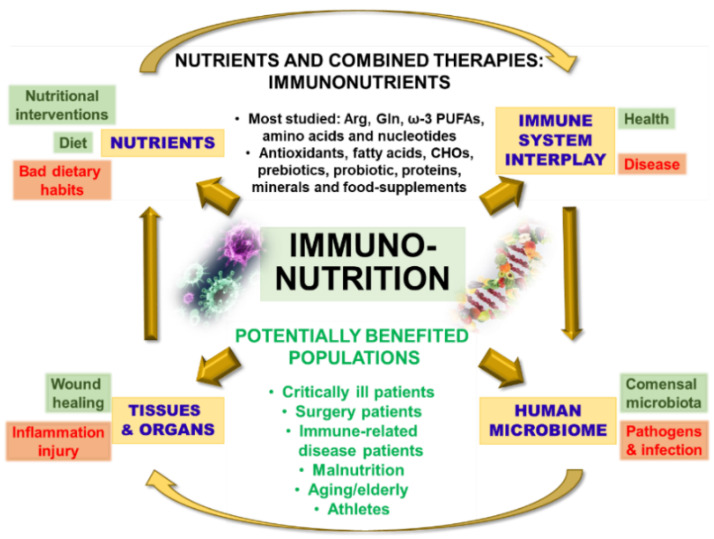
Immunonutrition overview. Basic pillars of immune-modulating nutrition. The term immunonutrition is mainly powered by four interconnected concepts: immune system, nutrition, body organ metabolism, and the human microbiome. These relationships pave the way for a wide and complex multidisciplinary topic with potential applications. The immune system comprises an intricate network of mechanisms, signals, cells, organs, and tissues. It is extraordinarily variable between subjects and when it fails, it can shift the fine line between health and disease. Bad dietary habits can lead to the poor nutritional status of the subject and consequently to severe pathologies. Nutritional interventions are needed at this point. Single nutrients or combined therapies, all of them designated as immunonutrients, are considered key tools. The most studied are arginine (Arg), glutamine (Gln), n-3 fatty acids (ω-3 PUFAs), amino acids, and nucleotides. Antioxidants, fatty acids, carbohydrates (CHO), prebiotics, probiotics, proteins, minerals, and food supplements are other employed immunonutrients. Deficiency of certain nutrients leads to compromised immunity and a high predisposition to infections. The immune system fights against harmful agents such as bacterial antigens supported by its ability to distinguish between self and nonself molecules. This includes foreign components and commensal microbiota. After infection, innate and adaptive immune responses occur, generally through inflammatory processes and metabolic changes. These events can trigger tissue injury, as well as undernutrition or malnutrition. An inadequate intake of macro- and micronutrients negatively affects the immune system. Nutrients help repair tissues (wound healing), modulate cytokine release, or protect organs from free radicals or oxidants. Immunonutrition provides potential benefits to varied and numerous populations: from critically ill patients to professional athletes, and including patients undergoing surgery, subjects with immuno-related diseases, people at potential risk of malnutrition, or the elderly.

**Figure 2 biomedicines-09-00519-f002:**
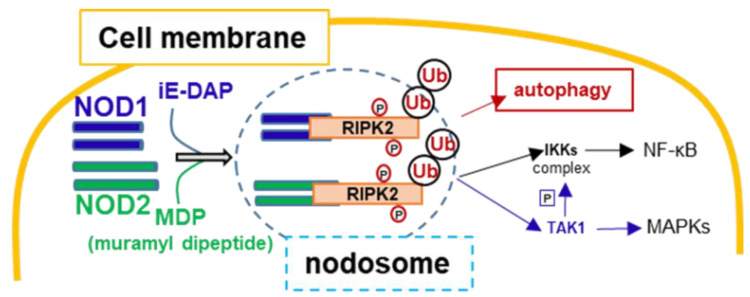
NOD1/NOD2 (*nodosome*) activation and signaling. NOD1 or NOD2 dimerize after MAMPs/PAMPs binding, recruiting the kinase RIPK2 that becomes autophosphorylated and polyubiquitinated. This *nodosome* promotes additional intracellular signaling via TAK1/IKK activation that results in the activation of Mitogen-Activated Protein Kinases (MAPKs) and NF-κB translocation to the nucleus.

**Figure 3 biomedicines-09-00519-f003:**
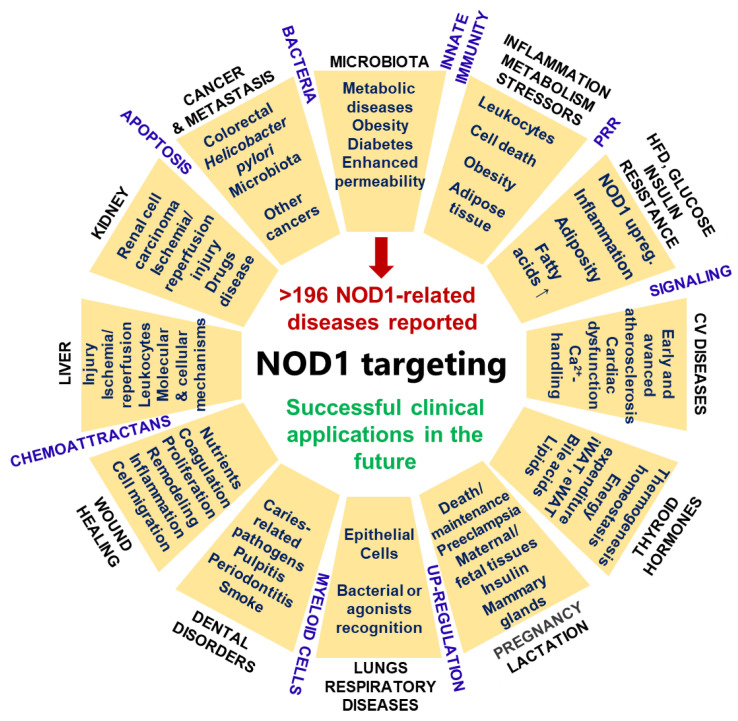
Health or disease: a matter of the well-known PRR NOD1 with possibilities of taking advantage of alliances with immunonutrition. NOD1, a cytoplasmic PRR of the innate immunity, is expressed in nonhematopoietic cells (such as endothelia) and in hematopoietic and immune cells (e.g., monocytes/macrophages, neutrophils, NK cells, lymphocytes). PRRs recognize molecular patterns associated with microbes or pathogens (MAMPs or PAMPs) and become activated. Impairments in this activation lead to the development of autoimmunity and other diseases. Components of the bacterial cell wall (peptidoglycan (PGN) and murein), are recognized and activate NOD1. γ-D-glutamyl-*meso*-diaminopimelic acid (iE-DAP) is sufficient to accomplish NOD1 activation. iE-DAP is mainly present in Gram-negative bacteria, which constitute the main microorganisms in the human gut. There exist at least 196 reported diseases related to NOD1, ranging from several cancers or neoplasms to metabolic, inflammatory, immune, and infectious diseases, as summarized in this figure. NOD1 is overexpressed and activated under these pathological conditions. Therefore, NOD1-related diseases connected with nutrient metabolism, deficiencies, or disorders can benefit from carefully studied immunonutrition approaches and clinical applications. *Blue letters*, events, and pathways related to NOD1 signaling.

**Figure 4 biomedicines-09-00519-f004:**
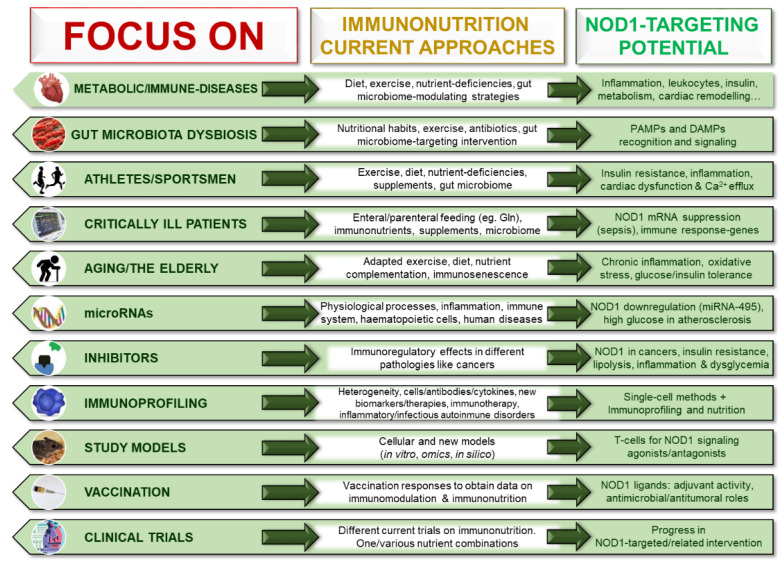
Looking to the future: promising novel NOD1–immunonutrition perspectives. Integrated, interdisciplinary, and personalized strategies are considered the future in the fight against today’s global diseases. This figure collects different fields related to existing immunonutrition strategies and that could exploit the potential of a dual approach with interventions targeting NOD1. Metabolic and immunological disorders (such as obesity or diabetes), content and function of the intestinal microbiota, athletes or professional athletes (suffering from respiratory diseases, infections or gastrointestinal disorders), critically ill patients, the elderly, mechanisms of aging/senescence, microRNAs, inhibitors immune system and immune checkpoint inhibitors, immunoprofiling, biological or nonbiological study models (highlighting cell and animal models), vaccination flow diagram and ongoing clinical trials on immunonutrition (using various combinations of immunonutrients) are discussed here. NOD1 has almost undiscovered potential as a therapeutic target that should be used against human disorders. New treatments can complement or improve existing therapeutic and nutritional approaches.

## Data Availability

Not applicable.

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
