# Peer review of "NOD1-Targeted Immunonutrition Approaches: On the Way from Disease to Health"

_biomedicines, 2021, doi:10.3390/biomedicines9050519_

Round 1

Reviewer 1 Report

The authors described that the immunonutrition approaches targeting NOD1 are potentially useful for intervention of various diseases including infection, autoimmune diseases, metabolic diseases, aging, and cancer. This concept of the approaches is interesting.

I have some questions and suggestions.

1. In page4, the authors explained that NOD1 is one of pattern-recognition receptors (PRRs) expressed in immune cells. Are there other PRRs than NOD1? What is the character of each PRR?

2. It is a little difficult to understand the basic biological functions of NOD1. I think it is easier to understand if there is a figure indicating some functions of NOD1.

3. In page5, the authors described that the myeloid F4/80+ population appears to be susceptible to the immunonutritional-driven polarization by dietary protease inhibitors. Is this description for mouse or human, or both? F4/80+ cell population is different between mouse and human.

4. In page6, the authors mentioned that NOD1 has been connected with cell death induction. How is NOD1 connected with cell death induction? The more detailed mechanism should be written.

5. In page5-10, the authors described the relationships of NOD1 with various diseases. The description is not so well-organized. The text should be divide by the types of diseases and each part should be entitled.

Author Response

REVIEWER 1

The authors described that the immunonutrition approaches targeting NOD1 are potentially useful for intervention of various diseases including infection, autoimmune diseases, metabolic diseases, aging, and cancer. This concept of the approaches is interesting.

A/ We thank the Reviewer for the effort in improving the quality of the manuscript. We included all suggestions in the new version of the work. Also, English has been revised in depth, following the comments of the Reviewers.

I have some questions and suggestions.

1.In page4, the authors explained that NOD1 is one of pattern-recognition receptors (PRRs) expressed in immune cells. Are there other PRRs than NOD1? What is the character of each PRR?

A/ We have reinforced the references regarding the PRR. In the review, we have included comments on the role of NOD2  in the context of inflammation and sensing specific microbial motifs. New references have been included to provide readers with appropriate information (refs: 23, 26-31, page 5; 37-40, pages 5; 84, 86 and 88, page 8).

2.It is a little difficult to understand the basic biological functions of NOD1. I think it is easier to understand if there is a figure indicating some functions of NOD1.

A/ We thank the Reviewer for this comment. In agreement of these comments, we have introduced a new figure 2 describing the main biological signaling of NOD1, in the context of the nodosome structure. We think that this new figure may be useful for general readers of the review.

3.In page5, the authors described that the myeloid F4/80+ population appears to be susceptible to the immunonutritional-driven polarization by dietary protease inhibitors. Is this description for mouse or human, or both? F4/80+ cell population is different between mouse and human.

A/ The Reviewer is correct in the comment. Since the sentence refers to both human and murine macrophages, we have corrected the myeloid markers for each organism.

4.In page6, the authors mentioned that NOD1 has been connected with cell death induction. How is NOD1 connected with cell death induction? The more detailed mechanism should be written.

A/ The mechanisms of NOD1-dependent cell death have been described by different authors. One of the best mechanisms is through caspase 8 activation. This has been implemented in the text and with new references reporting the main mechanisms involved in this cell death dependent NOD1 activation (References 52-55, page 6).

5.In page5-10, the authors described the relationships of NOD1 with various diseases. The description is not so well-organized. The text should be divide by the types of diseases and each part should be entitled.

A/ We thank the Reviewer for this comment. Our aim was to provide data on current research in immunometabolism, microbiota and the role of NOD1 in several diseases. Also, we have included relevant clinical trials focused on immunometabolic intervention as a therapeutic option. Accordingly, we have introduced sub-sections specifics for each disease.

Reviewer 2 Report

In this manuscript by Fernández-García et al., the authors have provided a collective information on NOD1-targeted immunonutrition approaches. This is an interesting article for the readers of specific field. However, the manuscript should be extensively edited before reconsidering for publication.

  1. The manuscript has collection of information without connectivity between paragraphs or sentences. The authors should use subheadings to improve readability. English language editing will help.
  2. "A focus on future strategies" section should be reduced significantly.
  3. The authors should elaborate the structure and induced proximity model of NOD1/2 activation for better understanding.
  4. A section on the regulation of NOD1/2 by ubiquitination should be included.
  5. The authors should provide a detailed section on Nodosomes.
  6. Elaborate the clinical significance of NOD1 with relevant ongoing/completed clinical trials as well as appropriate references.

Author Response

In this manuscript by Fernández-García et al., the authors have provided a collective information on NOD1-targeted immunonutrition approaches. This is an interesting article for the readers of specific field. However, the manuscript should be extensively edited before reconsidering for publication. The manuscript has collection of information without connectivity between paragraphs or sentences. The authors should use subheadings to improve readability. English language editing will help.

A/ We would like to thank BIOMEDICINES Editors and Reviewers for their positive assessment of our manuscript and invitation to submit a revision. We found the comments very helpful and addressed all of them. We believe these modifications have significantly improved the quality of our manuscript, including an extensive English revision.

1."A focus on future strategies" section should be reduced significantly.

A/ Following the Reviewer's suggestion we have significantly shortened this section, keeping the key issues for future strategies based on immunonutrition interventions. We hope this section is appropriate to provide information to readers of this review.

2. The authors should elaborate the structure and induced proximity model of NOD1/2 activation for better understanding. A section on the regulation of NOD1/2 by ubiquitination should be included. The authors should provide a detailed section on Nodosomes.

A/ Following this suggestion, we have introduced a new figure (figure 2) that provides quick information on how to activate NOD1/NOD2. In addition, we have included in this figure references on polyubiquitination (intended for those readers who are interested in the dynamics of NOD1/2-RIPK2 signaling and the constitution of the nodosome structure). We sincerely thank the Reviewer for this requirement that was not covered in the previous version. References 28, 31 and 37-40 can help to obtain additional in-depth information on these important topics.

3. Elaborate the clinical significance of NOD1 with relevant ongoing/completed clinical trials as well as appropriate references.

A/ We reviewed information from immunonutrition clinical trials. On page 6, section 3, we provide the link to NOD1-associated diseases. Additionally, page 13 provides current clinical trials focused on immunonutrition. All of these essays have the corresponding reference to be used by the readers in the follow-up of the current essays. We believe that we have provided the most relevant information on these clinical trials.

Round 2

Reviewer 2 Report

The authors have addressed all the comments. The manuscript was improved significant by revision and can be accepted for publication.